# Biological Biomarkers of Response and Resistance to Immune Checkpoint Inhibitors in Renal Cell Carcinoma

**DOI:** 10.3390/cancers15123159

**Published:** 2023-06-12

**Authors:** Claire Masson, Jonathan Thouvenin, Philippe Boudier, Denis Maillet, Sabine Kuchler-Bopp, Philippe Barthélémy, Thierry Massfelder

**Affiliations:** 1Regenerative NanoMedicine, Centre de Recherche en Biomédecine de Strasbourg, Fédération de Médecine Translationnelle de Strasbourg (FMTS), UMR_S U1260 INSERM and University of Strasbourg, 67085 Strasbourg, France; claire.masson@etu.unistra.fr (C.M.); kuchler@unistra.fr (S.K.-B.); 2Medical Oncology Department, Hospices Civils de Lyon, Hôpital Lyon Sud, 69310 Pierre-Bénite, France; jonathan.thouvenin@chu-lyon.fr (J.T.); denis.maillet@chu-lyon.fr (D.M.); 3Medical Oncology Department, Institut de Cancérologie Strasbourg Europe, 67200 Strasbourg, France; p.boudier@icans.eu

**Keywords:** clear cell renal cell carcinoma, immune checkpoint inhibitors, immunotherapy, biomarkers, resistance

## Abstract

**Simple Summary:**

A several immunotherapy-based combinations regimen improved overall survival, and became the new first-line standard of care for metastatic clear cell renal cell carcinoma (mccRCC) patients. However, a subset of patients, approximatively 30 to 60%, are refractory to these therapies or develop a secondary acquired resistance. Currently no validated biomarkers are available in routine practice to adapt treatment strategies. Here, we summarize the potential biological biomarkers for immune checkpoint inhibitors that may have arisen from the studies of immune checkpoints.

**Abstract:**

Renal cell carcinoma (RCC) represents around 2% of cancer-related deaths worldwide per year. RCC is an immunogenic malignancy, and treatment of metastatic RCC (mRCC) has greatly improved since the advent of the new immunotherapy agents, including immune checkpoint inhibitors (ICIs). However, it should be stressed that a large proportion of patients does not respond to these therapies. There is thus an urgent need to identify predictive biomarkers of efficacy or resistance associated with ICIs or ICI/Tyrosine kinase inhibitor (TKI) combinations; this is a major challenge to achieve precision medicine for mRCC in routine practice. To identify potential biomarkers, it is necessary to improve our knowledge on the biology of immune checkpoints. A lot of efforts have been made over the last decade in the field of immuno-oncology. We summarize here the main data obtained in this field when considering mRCC. As for clinical biomarkers, clinician and scientific experts of the domain are facing difficulties in identifying such molecular entities, probably due to the complexity of immuno-oncology and the constant adaptation of tumor cells to their changing environment.

## 1. Introduction

Kidney cancer is the seventh leading cancer worldwide, accounting for around 3% of all cancers, with more than 430,000 cases and 179,000 deaths per year [1,2]. Clear cell renal cell carcinoma (ccRCC) is the main subtype of kidney cancer, representing 70% of all cases, followed by non-clear cell RCC, including different subtypes, mainly papillary subtype and chromophobe RCC. RCC are mostly diagnosed at a localized stage, though approximately 20–40% will develop metastasis after nephrectomy, and nearly 35% patients will present metastatic disease at diagnosis [3,4]. Despite the landmark advances in advanced RCC over the last decade, with overall survival reaching 47 months, we still have a need for biomarkers to guide our treatment strategies [5,6]. Thus, for example, patients with clinically “similar” tumors may have significant different therapeutic responses to a same-combination regimen. Consequently, one of the major challenges in mRCC is to define and validate clinical and molecular predictive biomarkers for patient selection and to define the appropriate treatment.

Antiangiogenic small-molecule tyrosine kinase inhibitors (TKIs) have been the therapies of reference for mRCC for years. These therapeutic molecules arose from the knowledge of the von Hippel-Lindau (VHL)/Hypoxia-induced factors (HIF) system, targeting, among others, VEGF, PDGF, and FGF signaling pathways such as Sunitinib, Pazopanib, Axitinib, and Tivozanib [7,8,9,10], or the VEGF-targeted antibody bevacizumab [11] and the mammalian target of rapamycin (mTOR) inhibitors Everolimus and Temsirolimus [12]. However, although these compounds have shown significant anti-tumor activity, with prolonged responses and survival, mRCC remains largely an incurable disease. With better knowledge of the mechanism of resistance to first-generation TKIs, new targeted compounds have been developed to overcome these mechanisms of resistance for mRCC, such as Cabozantinib (MET, VEGFR2, AXL and RET inhibitor), Lenvatinib (FGFR inhibitor) or Savolitinib (MET specific inhibitor). Savolitinib is still in development, particularly for the papillary subtype harboring C-MET alterations. Belzutifan, an HIF2α inhibitor, is the last compound currently in development in previous heavily pretreated patients (MK 6482 005 trial); this later compound is furthermore evaluated in combination with Lenvatinib compared to Cabozantinib in the clinical phase III LITESPARK-011 trial as first-line or second-line therapy [13], or in combination with Pembrolizumab in the adjuvant setting.

The more recent development of immune checkpoint inhibitors (ICIs) has revolutionized the therapeutic approaches of multiple cancer types, including ccRCC. Immune checkpoints are key regulators of the immune system preventing autoimmune responses. Many stimulatory or inhibitory checkpoint molecules do exist that are expressed on T cells, macrophages and NK cells, such as CD27, CD40, CD122 or CD276, B- and T-lymphocyte attenuator 4 (BTLA4) and the well-known cytotoxic T-lymphocyte-associated protein-4 (CTLA-4), Programmed death-1 (PD-1), Lymphocyte-activation gene 3 (LAG) and T cell immunoglobulin and mucin domain 3 (TIM-3). Immune checkpoint molecules may also be expressed by tumor cells to promote an immunosuppressive microenvironment that ultimately leads to tumor growth [14], and their blockade allows the immune system to retrieve the ability to kill tumor cells.

ICIs targeting PD-1 (Nivolumab) and CTLA4 (Ipilimumab) have been approved for RCC therapies. These agents have improved overall survival as monotherapy, as well as in combination with other ICIs or anti-angiogenic therapeutic compounds. However, most patients will progress, and there is still a lack of understanding of which patients will benefit from these therapies. A complete response occurs only in approximatively 10–17% of patients with anti PD-1/anti-CTLA4 combinations as first-line therapies. In addition, 5–20% of patients will be refractory to ICI/TKI-based combinations. Moreover, toxicity management remains a challenge in routine practice, as some patients experience severe immune mediated adverse events leading to treatment interruption or discontinuation. We recently discussed all these later results [15].

The development of predictive biomarkers of efficacy and toxicity is a research priority, and is urgently needed to optimize our treatment strategies in the immunotherapy era. Biomarker identification may help in the future to answer the remaining unresolved questions concerning the optimal duration of treatment, the best therapeutic sequence, and the role of surgery to remove residual disease. The need to develop additional ICIs for patients who are initially resistant to treatments will also be important in the near future.

Biomarkers will also be of major interest in the adjuvant setting. The identification of the best candidate for adjuvant therapy is crucial, as only Pembrolizumab may be considered an ICI option for a selected population; the results of the KEYNOTE-564 phase 3 trial support the use of Pembrolizumab as adjuvant immunotherapy in patients with intermediate-to-high or high risk of RCC disease recurrence [16]. However, results from the CheckMate 914 and IMotion 010 trials have been conflicting. Additional investigations are obviously needed, and probably by examining specific sub-groups of patients.

In this review we will focus on the potential biological biomarkers for ICI therapies and combination therapies in RCC arising from a better knowledge of immune microenvironment.

## 2. The Biology of Immune Checkpoints

To identify biomarkers for ICIs response, it is important to have good notions in the biology of immune checkpoints. This is a complex area of research that is in constant evolution, with the identification of signaling pathways stimulated by these receptor/ligand couples, and the discovery of new immune checkpoints playing roles in the response of tumor cells to ICIs. An uncontrolled immune response can induce autoimmune diseases, inflammatory tissue damage, chronic infection, and cancers; thus, maintaining immune homeostasis is an absolute necessity for host survival, and this is achieved by a careful balance between stimulatory and inhibitory immune signals, collectively referred to as immune checkpoints. Activated T cells are primary mediators of immune functions, and express multiple inhibitory immune checkpoint receptors molecules including CTLA-4, PD-1, LAG3, TIM-3, BTLA4, killer immunoglobulin-like receptors (KIR), the T-cell immunoglobulin and ITIM domain (TIGIT), tumor necrosis factor receptor (TNFR) superfamily, V-domain Ig suppressor of T cell activation (VISTA), and Indoleamine 2, 3-dioxygenase 1 (IDO1), which regulate the T cell responses to tumor antigens [17]. James Allison and Tasuku Honjo were at the origin of the discovery of the first two identified immune checkpoint molecules, CTLA-4 and PD-1, and received the Nobel Prize in 2018 as a recognition of their discovery. An interesting feature of these checkpoints receptor/ligand pairs molecules is that they use not only unique signaling pathways, but also pathways that are non-reduplicative (stimulation or inhibition), such as JAK/Stat, PI3K/Akt, MAPK and/or NF-kB pathways [18,19]. Consequently, there is an open rationale logic to target many of these immune checkpoints to increase anti-tumor activity.

Tumor cells have developed several strategies to exploit these checkpoints and circumvent the host immune defenses. In other words, cancer cells can mimic the ligands of immune checkpoints to evade immune surveillance [20]. These ligands are normally expressed on antigen-presenting cells (APC), a heterogeneous group of immune cells (dendritic cells, macrophages, B cells…) that mediate the cellular immune response by processing and presenting antigens for recognition by T cells [21]. In addition to inhibitory immune checkpoints, stimulatory immune checkpoints are also promising targets for immune therapy. These include 4-1BB (a co-stimulatory member of the TNFR superfamily), glucocorticoid-induced TNFR-related (GITR, which also belongs to the TNFR superfamily), OX40, or CD134, inducible co-stimulator (ICOS, a member of the B7-CD28 superfamily), but also CD28, CD27, CD28H, CD30, CD40, CD122, and CD137 [22]. They all have their own ligand on APC (such as CD80, CD86 for CTLA-4 or PD-L1 and PD-L2 for PD-1), and tumor cells, depending on tumor types, may substitute them, thus using this immune checkpoint machinery to avoid cell killing. Stimulatory checkpoint pathways promote immune responses, while inhibitory checkpoint pathways inhibit immune responses. This allows cancer cells to escape capture by the immune system, and immune checkpoint therapy has been developed to counter this process. Some tumor types show low immunogenicity, and may not respond effectively to an immune checkpoint blockade. Some tumors are classified as ‘cold’ in terms of immunogenicity, with a microenvironment with T-Reg and immunosuppressive cells. There are now bispecific antibodies in development to shift these tumors to ‘hot’ tumors in terms of immunogenicity (such as for chromophobe RCC, which is a cold tumor). ccRCC is known to be an immunogenic malignancy, and this supports the potential therapeutic effect of ICIs in mccRCC, although there is still an important subset of patients who will not benefit from these therapies. Some patients initially respond to ICIs, but the presence of clones with low immunogenicity will consequently induce tumor relapses. For non-responder patients this can even stimulate tumor progression [23], which can be assimilated to tumor resistance. There are still other immune checkpoint molecules expressed on immune cells, such as for example B- and T-lymphocyte attenuator (BTLA or CD272, very similar to CTLA-4 and PD-1), adenosine 2A receptors (A2AR, a G protein-coupled receptor with a high affinity for adenosine), phosphate oxidase 2 (NOX2), and sialic acid binding Ig like lectin 7 (SIGLEC7). As evoked above, the choice of the best combination strategy should be driven by the knowledge of the similarities and differences in the signaling pathways of the different immune checkpoints expressed on the cell surface [18,19]. Finally, promising approaches for cancer treatment also include the combination of ICIs with other kinds of immune therapy, such as chimeric antigen receptor (CAR)-T cells (CAR-T), T-cell receptor (TCR)-T cells (TCR-T) or vaccines [24,25,26].

Although there are numerous excellent reviews on immune checkpoints and ICIs, it was important here to make this short update in this chapter. Indeed, we are still far from having gone around all the possibilities to inhibit these immune checkpoints and all the possible combinations.

## 3. Biological Biomarkers of ICIs Response

Predictive biomarkers of efficacy, resistance, and toxicity must meet the following 3 essential criteria: prospective validity; clinical utility; and feasibility. Several reviews have summarized potential biomarkers in various cancer types [15,22,27]. However, RCC is a particular subtype, as it is known to be very immunogenic. We tried here in this chapter to summarize data obtained in this field focusing on mRCC.

### 3.1. miRNA Signature

In their report, Incorvaia et al. hypothesized that a miRNA signature could be identified in the peripheral lymphocytes of mRCC patients treated with second line Nivolumab, which could be used to discriminate responder from non-responder patients [28]. It should be noted that no biological epigenetic biomarkers are currently used in the clinic. They enrolled 23 patients in a prospective study, and found a specific subset of miRNAs specifically induced in long-responder patients to Nivolumab (>18 months), i.e., miR-22, miR-24, miR-99a, miR-194, miR-214, miR-335, miR-339, and miR-708. Interestingly, as noted by the authors, these miRNAs have been shown to be involved in cell proliferation, cell cycle regulation, apoptosis, migration, and invasion of RCC. Obviously, this will have to go through thorough validation on much larger cohorts, and in first line with ICI/ICI or ICI/TKI combination (Table 1).

### 3.2. TDO and IDO1 Status

More recently, Sumitomo et al. aimed at investigating the involvement of tryptophan 2,3-dioxygenase (TDO) and of indoleamine 2,3-dioxygenase 1 (IDO1) in both cancer development and resistance to ICIs in patients with mRCC or non-metastatic disease [29]. IDO1 expression was shown earlier to be increased not only in RCC tumor cells, but also in the microenvironment of human RCC compared to normal kidney tissues [52]. IDO1 regulates the tryptophan-kynurenine (Kyn) pathway in advanced cancers, including mRCC, and thus appears as a key enzyme associated with immunomodulation and tumor immune evasion [53]. The interest of the authors was based on the negative impact observed in clinical trials using IDO1 inhibitors in combination with ICIs in patients with mRCC [54,55]. The mRCC cohort studied constituted 40 mRCC patients (36 mccRCC, 4 other mRCC) treated with Nivolumab only (*n* = 32), Nivolumab and Ipilimumab (*n* = 6), and TKIs and PD-L1 antibody (*n* = 2). They showed the expression of TDO rather than IDO1 in tumor cells, and its strong association with Kyn expression, but also with forkhead box P3 expression, as well as its association with tumor progression, with ICIs suggesting its potential as a predictive biomarker of resistance to ICIs in patients with mRCC. Again, these results will have to be validated in larger cohorts (Table 1).

### 3.3. Circulating Cytokine Levels

In their original study, dealing with the cytokine levels in the circulation of mRCC patients receiving either ICI or TKI therapy (Nivolumab, Cabozantinib, Nivolumab/Ipilimumab, Sunitinib, Lenvatinib/Everolimus, or Axitinib), Chehrazi-Raffle et al. prospectively enrolled 56 patients who were planned for treatment with either therapy [30]. This was the first evaluation of the potential relation between circulating cytokine levels (at baseline and on-treatment) and ICI response. A panel of 30 plasma cytokines (including IL-1RA, IL-1b, IL-2, IL-2R, IL-4, IL-5, IL-6, IL-7, IL-8, IL-10, IL-12, IL-13, IL-15, IL-17, Eotaxin, EGF, FGF, G-CSF, GM-CSF, IFN-α, IGN-γ, CXCL9, CXCL10, CCL2, CCL3, CCL4, RANTES, TNF-α, and VEGF) was quantified in the plasma of each patient. Clinical benefit was defined as CR, PR, or SD ≥6 months, and was shown to be similar for patients receiving either therapy. They report a clinical benefit for patients with lower levels of IL-1RA, IL-6, and G-CSF at pretreatment and of IL-12, IL-13, IFN-γ, GM-CSF, and VEGF during treatment. These results are interesting since circulating plasmatic cytokine levels can be easily measured; it remains to be determined whether they can be validated in other cohorts, and whether they can be adapted in the clinical settings. Indeed, in recent retrospective analyses of large phase II and phase III trials, elevated baseline serum IL-8 was found to be correlated with poor response and resistance to ICIs treatment in various cancer types, including RCC [56,57], but not in the above-mentioned study [30]. This apparent discrepancy raises the question of the potential role of this cytokine as a biological biomarker for ICI response (Table 1).

### 3.4. Circulating Tumor DNA

Circulating tumor DNA (ctDNA) has demonstrated potential as a surveillance biomarker for disease recurrence in a localized RCC setting [58]. Most patients with mRCC have detectable ctDNA, and their potential role as a predictive biomarker for response to ICIs was evoked a few years ago. This potential predictive role of ctDNA in ICIs response was assessed in two separate studies on 20 RCC (including 10 with mRCC) patients [31] and 14 mRCC patients [32], i.e., small-scale cohorts. In the first above-mentioned study, authors examined the predictive role of ctDNA in only four mRCC patients (on 10 available) who received first-line ICI therapy (Nivolumab/Ipilimumab combination). With their observations, they concluded that the levels of ctDNA could be an early predictor of treatment response to ICIs in patients with mRCC who receive ICI therapy, with an inverse association between the level of ctDNA and therapeutic response, i.e., patients with decreasing ctDNA levels had a better therapeutic response. Koh et al. report a similar conclusion [32] based on results suggesting a better PFS for mRCC patients with decreasing ctDNA mutant allele frequency compared to those with increasing mutant allele frequency. However, these results are exploratory and need prospective validation on large cohorts.

In their recent publication, Chehrazi-Raffle et al. evaluated whether targeted digital sequencing (TARDIS) may distinguish a partial response from a complete response) among patients with mRCC receiving ICIs [33]. In this pilot study, they analyzed 12 patients receiving either Nivolumab monotherapy (6/12) or Nivolumab plus Ipilimumab (6/12) therapy. They obtained peripheral blood at a single time point for ctDNA analysis, and used TARDIS to quantify the average variant allele fractions (VAFs) to determine (i) whether there is an association between VAFs and partial response or complete response, and (ii) whether VAFs were associated with disease progression. An average of 30 patient-specific mutations were incorporated in the ctDNA analysis, and they indeed showed that TARDIS accurately differentiated partial to complete responses and allowed to identify patients at risk of tumor progression. As acknowledged by authors, the results of this pilot study will need to be validated for decision-making as yes or no in terms of whether ICIs therapy should be continued. But interestingly these results go in the same sense than the ones cited above (Table 1).

### 3.5. CDKN2A Tumor Suppressor Gene Genetic Status

The impact of loss-of-function of the tumor suppressor gene *CDKN2A* (cyclin-dependent kinase inhibitor 2A, located at chromosome 9, p21.3) on response and survival in patients treated with ICIs was analyzed by Adib et al. in two independent large-scale cohorts [34]. The loss-of-function of this gene is frequent in ccRCC (1/3 of all cases). However, although the loss of chromosome 9p21 is associated with poorer patient survival, 9p21 loss of heterozygosity was shown by Baietti et al. [59] to not lead to decreased expression of CDKN2A, suggesting alternative mechanisms of 9p-mediated tumorigenesis; indeed, in this latter work, 9p21 loss was shown to relieve the *HOXB13* (homeobox gene 13) locus, promoting its expression, and thus tumor cell growth, which is associated with poorer survival of ccRCC patients. In their study, Adib et al. [34] included six different cancer types, including RCC, comprising 789 patients (56 with RCC) treated at the Dana-Farber Cancer Institute in Boston, USA, and 1250 patients (151 with RCC) at the Memorial Sloan Kettering Cancer Center, New York, USA, respectively. In both cohorts, *CDKN2A* genetic alterations were associated with poor response to ICI, either Nivolumab, Ipilimumab or both, and survival in patients with urothelial carcinoma, but no association was observed in the other cancer types, including RCC, which is reminiscent of the conclusion drawn by Baietti et al. [59].

More recently, Xu et al. [35] showed that mutated deficient CDKN2A in RCC patients is associated with sarcomatoid differentiation, tumor progression, poor prognosis and primary resistance to Sunitinib, and potential favorable responses to ICIs. The same observations were made with methylthioadenosine phosphorylase (MTAP), which is a key enzyme of the methionine remediation synthesis pathway, also located on chromosome 9p21, close to the *CDKN2A* locus, and thus the co-occurrence genomic alteration of *MTAP*/*CDKN2A* is frequent. In their large-scale and multicenter study, they enrolled 5307 RCC patients with genomic sequencing data from Chinese and Western cohorts (574 and 1170 Chinese RCC patients from the Fudan University Shanghai Cancer Center, FUSCC, and from the 3D cohort, respectively, and 3563 RCC samples from 17 independent Western cohorts). To reach these conclusions, they included all genomic alterations that led to a loss of protein function in the statistical analysis. Thus, and in contrast to the above-mentioned studies, CDKN2A deficiency was associated with a more favorable response to ICIs. Thus, even if the work of Xu et al. [35] included thousands of RCC patients, the precise role of *CDKN2A* mutations (and MTAP mutations) in the response of RCC patients to ICIs (Table 1) remains to be determined.

### 3.6. PBRM1 Tumor Suppressor Gene Genetic Status

*PBRM1* (Polybromo 1) gene alterations are associated with RCC, especially ccRCC (second-most frequent mutation in ccRCC after VHL inactivation) and non-papillary and ccPapillary RCC. PBRM1 protein is a subunit of the PBAF chromatin remodeling ATP-dependent complex necessary for ligand-dependent transcriptional activation by nuclear hormone receptors; it can act either as a tumor suppressor, for example in ccRCC, or oncogene, for example in prostate cancer. It should be stressed that following *VHL*, *PBRM1*, *SETD2*, *BAP1*, and *KDM5C* have been validated as common co-occurring gene mutations in ccRCC in multicenter studies [60]. In this latter study, results suggested that *PBRM1* mutation does not correlate with decreased survival, whereas BAP1 mutation seemed to predict poor outcome. However, this study only enrolled 20 patients with ccRCC. It should be noted that in a previous publication, Liu et al. [61] analyzed 3 independent patient cohorts and murine pre-clinical models to demonstrate that the loss of *PBRM1* is associated with a lesser immunogenic tumor microenvironment (decreased immune infiltrates), and consequently to resistance to ICI (anti-PD-1 treatment).

In their study, Miao et al. [36] performed whole-exome sequencing of mccRCC from 35 patients to identify genomic alterations that may correlate with response to anti-PD-1 monotherapy. They showed a clinical benefit with loss-of-function mutations of *PBRM1*, and confirmed this observation in another cohort with 63 mccRCC patients treated with ICIs alone (PD-1 or PD-L1 inhibitor) or in combination with anti-CTLA-4 inhibitor, arguing the fact that PBRM1 may appear as a biomarker for ICIs response in mccRCC. Later, Braun et al. [37], also studied whether alterations in the *PBRM1* gene were associated with clinical benefit to ICIs in a subset of another independent published cohort of 382 mccRCC patients (on 821 mccRCC patients initially) [62]. In this study, *PBRM1* mutations were identified in 55 of 189 mccRCC patients treated with Nivolumab (29%) and in 45 of 193 mccRCC patients treated with the TKI Everolimus (23%). They showed clinical benefit (OR and PFS) in the ICI-treated group, thus confirming the results obtained by Maio et al. [36], but not in the TKI-treated group. However, among limitations of the study on Braun et al. [37] there is the fact that the clinical benefit of *PBRM1* mutations on ICI response was observed in patients who received prior antiangiogenic therapy. Thus, as for *CDKN2A* status, the precise role of PBRM1 mutations as a biomarker for ICI response has still yet to be validated in additional large, randomized studies on mccRCC patients (Table 1).

Interestingly, more recently, Chen et al. [63], using bioinformatic methods, identified 37 immune-related genes associated with *PBRM1* mutations in ccRCC. In this original study, 520 ccRCC patients with survival information from TCGA (The Cancer Genome Atlas) database were analyzed. The height of these immune-related genes with high prognostic potential were identified and selected; these included *NPR3*, *MDK*, *IFNE*, *NTF4*, *PTGER1*, *GAL*, *FGF23*, and *CXCL13*. From these eight genes, authors developed a predictive model to assess the immune status, but also to predict the clinical outcomes of ccRCC patients, especially to ICIs. However, as stated by authors, the biological functions and molecular mechanisms of these genes in the immune response were not assessed, and thus remain largely unknown. In addition, it is a retrospective, and not a prospective study; thus, prospective clinical trials will be needed to validate the prediction performance of this model based on *PBRM1* genetic status.

### 3.7. Impact of AXL Expression

AXL is member of the TAM (TYRO3-AXL-MER) family of receptor tyrosine kinases. It regulates tumor cell proliferation, survival, migration, and angiogenesis, and also interacts with the microenvironment [64]. AXL has been associated with poor prognosis in many cancer types, including RCC and resistance to therapies [64]. In addition, its expression is also associated with aggravated clinical outcome in mccRCC treated with TKIs [65]. Various cancer models have linked AXL with immune suppression. In their recent study, Terry et al. explored the role of AXL in intrinsic and acquired resistance to the anti-PD-1 therapy by examining 316 mCCRCC patients who received Nivolumab after failure to anti-angiogenic therapies [38]. They showed that a high level of AXL expression in tumor cells is associated with lower response rates to the anti-PD-1 therapy and shorter progression-free survival. Interestingly, they showed that AXL expression was highly associated with PD-L1 expression in tumors, and more especially in tumors harboring VHL inactivation. Consequently, the worse overall survival was observed in patients displaying concomitant PD-L1 expression and high AXL expression. From this study, AXL appears as a biomarker of resistance to anti-PD-1 therapy. However, since this is the first demonstration of such a role in mccRCC, it will also need to be confirmed by additional studies in patients (Table 1).

### 3.8. CTLA-4 Genetic Status

Klümper et al. recently assessed the potential role of *CTLA4* (CTLA-4 encoding gene) promoter methylation as a biological predictive biomarker in response to anti-PD-1 ICI therapy in mRCC [39] (Table 1). This was based on their previous studies, showing that the methylation status of the *CTLA4* gene predicts response to both anti-PD-1 and anti-CTLA-4 targeted ICB as well as anti-CTLA-4 monotherapy in patients with melanoma [66,67]. They showed, through the analysis of three different cohorts of patients (533 cases from TCGA database, 116 patients not receiving ICI and 71 patients who received either anti-PD-1 monotherapy second-line or later post-TKI—*n* = 46, or first-line anti-PD-1-based combination therapy—*n* = 25), that *CTLA4* promoter hypomethylation was an independent predictor of improved outcome (PFS and OS) in ICI-treated ccRCC. Thus, the epigenetic regulation of the *CTLA4* gene might be promising as a predictive biological biomarker for ICI + ICI combination in patients with mRCC.

### 3.9. Soluble Molecules

Interestingly, more recently, a plasma proteomic profile of patients with mRCC treated with Nivolumab was performed by Simonetti et al. to identify soluble molecules potentially associated with clinical outcome [40]. They analyzed 507 soluble molecules in the pre-treatment plasma of 16 patients with mRCC. Among these 507 molecules, 12 were significantly elevated in non-responders compared to responders, and after adjustment, the only molecule retained was the nuclear factor kappa-Β ligand (RANKL), which was further validated as indeed being overexpressed in non-responder patients. As proof of concept, patients with low RANKL levels had significant improvements in PFS. RANKL thus appears as an independent biological biomarker in mRCC patients treated with Nivolumab. Again, however, this analysis was performed on a small-scale cohort, and will need validation on larger cohorts or clinical trials (Table 1).

### 3.10. Tumor Content of T Cells

In their recent retrospective study on 24 patients treated with Nivolumab and Ipilimumab, Kim et al. analyzed, using multiplexed immunohistochemistry, the tumor content of T cell subsets (CD8+ cytotoxic, Foxp3- CD4+ helper or regulatory T cells), B cells, macrophages (CD68+ CD206- M1, CD68+ CD206+ M2), and dendritic cells [41]. From this study, they concluded that Foxp3- CD4+ helper T cells, M1 macrophages, and CD137+ CD8+ T cells are potential predictive biomarkers and treatment targets to PD-1 and CTLA-4 inhibitors (Table 1).

### 3.11. Soluble Immune Checkpoint-Related Proteins

In their study, Wang et al. [42] investigated the potential prognostic role of soluble immune checkpoint-related proteins in ccRCC patients. The circulating levels of a large panel of immune checkpoint-related proteins (including soluble BTLA, PD-1, PD-L1, PD-L2, CTLA-4, TIMP-3, LAG-3…) were measured using a multiplex Luminex assay in 182 ccRCC patients. Authors then analyzed the association of these levels with the risk of recurrence and death. Tumor gene expression was also analyzed in 47 patients and 533 primary ccRCC tumors from the TCGA database. They showed that (i) soluble TIM-3 and soluble LAG-3 were significantly associated with advanced disease, (ii) soluble PD-L2 was associated with recurrence and (iii) soluble BTLA and soluble TIM-3 were associated with decreased survival. Although confirmation will be needed, these results argue favorably the fact that soluble proteins may have prognostic value in ccRCC (Table 1).

### 3.12. Microbiota

It has been well described that the composition of the gut microbiota affects anti-tumor immune responses, and pre-clinical and clinical outcomes following ICI therapy. In RCC, this was first evidenced by the fact that the use of antibiotics, which alter the gut microbiota, negatively affects the effectiveness of ICIs [68,69,70]. Subsequently, the stool bacteria composition was also shown to be affected by TKIs prior to ICIs [43], and temporal changes were also observed in microbiome composition over the course of ICI therapy with a consequence on treatment outcome [44,71]. In this later report, authors identified over-represented bacteria such as *Prevotella copri*, *Bifidobacterium adolescentis*, *Faecalibacterium*, and *Akkermansia muciniphila* in patients who experienced better outcomes; interestingly, some of these bacterial strains were shown to be associated with an increase in intratumor CD8+ T cells in other malignancies [72,73]. Similarly, in mRCC patients, high baseline diversity and *Akkermansia muciniphila* abundance correlated with favorable outcomes, while higher abundance of *Hungatella hathewayi* or *Clostridium clostridioforme*, for example, was seen in non-responders to Nivolumab [43]. The question of the benefit of fecal microbiota transplantation of course arose from all these studies for patients with urological malignancies treated with ICIs and that are resistant to therapies. Clinical trials are ongoing to answer this question [74]. It is of importance to cite the MITRE trial protocol, which aims at prospectively evaluating the microbiome as a biomarker of efficacy and toxicity in large-scale melanoma, RCC and non-small cell lung cancer patients receiving ICIs [45]. A microbiome signature is expected to be identified after extended analysis of stool samples, thereafter allowing manipulation of the microbiota composition for a better treatment outcome. A recent interesting review on the association of the human microbiome with clinical outcomes in RCC has been published by Piao et al. [75] (Table 1).

As antibiotics, proton pump inhibitors (PPIs) are also known to affect gut microbiota. In their recent study, Mollica V et al. [46] evaluated the impact of PPIs on outcomes in 2 cohorts of mRCC patients from five international centers receiving immunotherapy: a first cohort, 62 patients (25 receiving PPIs), receiving Ipilimumab and Nivolumab for first-line treatment, and the second cohort, 156 patients (88 receiving PPIs), receiving Nivolumab for second-line or third-line treatment. In this study, however, in neither cohort the concomitant use of PPIs affected survival outcomes on ICIs. Such results will also need to be validated in additional cohorts (Table 1).

### 3.13. ccRCC Molecular Subgroups

In 2015, 4 ccRCC (ccRCC 1-4) molecular subtypes were identified that are predictive of Sunitinib response in mRCC patients [76]. More recently, the 35-gene expression profile identifying these four groups was investigated in the BIONIKK phase 2 trial, which enrolled 202 patients to evaluate treatment efficacy and tolerability of Nivolumab alone, the combination Nivolumab/Ipilimumab, and VEGFR-TKIs in patients according to these 4 molecular groups [47]. The main result of this trial was the demonstration that patients could be selected according to this molecular signature to benefit from the more active treatment among the 3 tested (Table 1).

In 2020, the IMmotion 150 phase 2 randomized trial showed improved PFS for the Atezolizumab/Bevacizumab (ICI/VEGF inhibitor) combination vs. Sunitinib in previously untreated patients with mRCC expressing PD-L1 (2.8 months in all patients vs. 3.5 months in patients with PD-L1-positive tumors) [48]. This positive and encouraging result led to the ImMotion151phase 3 trial, which enrolled 915 patients with mRCC. The results of this trial were reported in 2022 [49]. As specified above, the primary endpoint, PFS, was met at intermediate analysis. However, no improvement in OS was observed with the Atezolizumab/Bevacizumab combination at the final analysis. A previously reported detailed exploratory biomarker analysis using seven molecular clusters was also performed during this trial [77]. This analysis showed improved median OS in patients whose tumors were characterized by T-effector/proliferative, proliferative, or small nucleolar RNA transcriptomic profiles, providing insight into which patients may benefit from either therapy (Table 1).

### 3.14. The Tumor Microenvironment

Finally—and far from being the last biological feature—is the place of the tumor microenvironment in ICI response. When considering the microenvironment, the well-known intra-tumoral (and inter-tumoral) heterogeneity of kidney cancer naturally comes to mind. Recent data demonstrate that the intra-tumoral heterogeneity (including highly aggressive sarcomatoid and rhabdoid features) influences the success of immunotherapy. Particularly, the acquired resistance to immunotherapy through the heterogeneity of neoantigens, the presentation of the antigen, and the interferon signaling were reported [78,79]. Data obtained in experimental models and in patients have shown that tumors with high intra-tumoral heterogeneity are less susceptible to respond to ICIs than more homogeneous tumors [80,81,82]. In their study, Zhang et al. performed single-cell RNA sequencing and developed benign and malignant cell atlases for RCC, providing insight into the possible cell of origin for the different RCC sub-types [50]. Among other interesting results they obtained was the observation that tumor epithelia promote infiltration of immune cells, which may explain why ccRCC tumors respond to ICIs, although they are known to have a low neoantigen burden. In addition, Bi et al. performed a single-cell transcriptomic analysis of both cancer and immune cells from mRCC patients before and after they received ICI (anti-PD-1 antibody Nivolumab) to study whether immunotherapy may modify these cells, and thus the microenvironment [51]. They showed that these cells are indeed reprogrammed after ICI treatment, and consequently the microenvironment, ultimately driving resistance to ICI. Interestingly, they also demonstrate that immune evasion is associated with *PBRM1* mutation, which is reminiscent of what we described above for this tumor suppressor (please refer to Section 3.6) (Table 1). Finally, resistance to immunotherapy through intra-tumoral heterogeneity could be due to different mechanisms; on one hand, more elevated intra-tumoral heterogeneity is associated with a more elevated risk to have a resistant clone, while on the other, some clones may not trigger an effective immune response.

## 4. Conclusions and Perspectives

Taking all of this together, although a lot of efforts have been made over the last few years to identify potential predictive biological biomarkers for ICI therapies, no one entered routine practice. None of these potential candidates have been prospectively validated in large cohorts or independent studies. Many works are still to be done in this field, and it is quite reminiscent of what we already observed for TKIs’ therapeutic response and resistance, i.e., no validated biomarkers so far.

Clinical rials assessing ICIs in different tumor types should systematically add translational research to make it easier and more efficient to identify potential predictive biomarkers. This is a rapidly expanding field, however, and many immune checkpoint molecules have already been identified, with more undoubtedly to come. This will also involve identifying the signaling pathways used by these receptor/ligand pairs, making it possible to avoid redundancies, and especially to designate the best combinations of inhibitors. Another field of investigation is also the identification of new targets expressed on tumor cells and inter-heterogeneity encountered in mRCC, which remains challenging.

This is a very complex area of research related to the biology of primary and secondary tumors and the immune system, which most likely vary from one individual to another, both spatially and temporally. Innovative materials and new experimental technologies are evolving rapidly, and suggest the possibility of hopefully identifying such biomarkers in the near future, at least for some patient populations.

## Figures and Tables

**Table 1 cancers-15-03159-t001:** Summary of the main findings for all biological biomarkers of response and resistance to ICIs.

Biomarker Studied/Alteration Reference(s)	Number of Patients	RCC Stage	ICI Treatment and Outcome
Lymphocyte miRNA signature			
[28]	23	mccRCC (*n* = 23)	Nivolumab Subset of miRNAs specifically induced in long-responder patients
Status of tryptophan 2,3-dioxygenase (TDO) and of indoleamine 2,3-dioxygenase 1 (IDO1)			
[29]	40	mccRCC (*n* = 36) mRCC (Other than ccRCC) (*n* = 4)	Nivolumab (*n* = 32) Nivolumab + Ipilimumab (*n* = 6) TKIs + PD-L1 antibody (*n* = 2) Expression of TDO, but not of IDO1 in tumor cells is strongly associated with tumor progression with ICIs suggesting its potential as a predictive biomarker of resistance to ICIs
Circulating cytokine levels			
[30]	56	mccRCC (*n* = 47) mRCC (Other than ccRCC) (*n* = 9)	Nivolumab (*n* = 25) Cabozantinib (*n* = 10) Nivolumab + Ipilimumab (*n* = 8) Sunitinib (*n* = 7) Lenvatinib + Everolimus (*n* = 5) Axitinib (*n* = 1) Clinical benefit for patients with lower levels of IL-1RA, IL-6, and G-CSF at pretreatment and of IL-12, IL-13, IFN-γ, GM-CSF, and VEGF during treatment
Circulating tumor DNA			
[31]	20	mRCC (*n* = 10) non mRCC (*n* = 10)	Nivolumab + Ipilimumab (in 4 mRCC only) Levels of ctDNA could be an early negative predictor of treatment response to ICIs in patients with mRCC who receive ICI therapy
[32]	14	mccRCC (*n* = 14)	Nivolumab (*n* = 10) Nivolumab+ Ipilimumab (*n* = 4) Better PFS for mRCC patients with decreasing ctDNA mutant allele frequency compared to those with increasing mutant allele frequency
[33]	12	mccRCC (*n* = 12)	Nivolumab (*n* = 6) Nivolumab + Ipilimumab (*n* = 6) The use of targeted digital sequencing (TARDIS) differentiates partial to complete response and allowed to identify patients at risk for tumor progression
CDKN2A tumor suppressor gene loss of function			
[34]	56 (789 multiple cancer types)	mRCC	Nivolumab Ipilimumab Nivolumab + Ipilimumab
	151 (1250 multiple cancer types)	mRCC	CDKN2A genetic alterations were not associated with response to ICIs in mRCC
[35]	5307 from Chinese and Western cohorts	mRCC	Anti-angiogenic targeted therapy ICIs Genomic alteration of *MTAP*/*CDKN2A* significantly correlates with sarcomatoid differentiation in RCC and predicts aggressive progression, poor prognosis, primary resistance to targeted therapy, and potential favorable responses to ICIs ≠ Results from [34]
PBRM1 tumor suppressor gene genetic alterations			
[36]	35	mccRCC	Nivolumab
	63	mccRCC	Nivolumab Atezolizumab Clinical benefit to ICIs treatment with loss-of-function mutations of PBRM1 in both cohorts
[37]	382	mccRCC	Nivolumab (*n* = 189) Everolimus (*n* = 193) Clinical benefit (OR and PFS) in the ICI-treated group (confirming results in [36]), but not in the TKI-treated group
Impact of AXL expression			
[38]	316	mccRCC	Nivolumab after failure to anti-angiogenic therapies AXL expression in tumor cells is associated with lower response rates to the ICI therapy and shorter PFS. AXL expression is highly associated with PD-L1 expression in tumor (more especially in tumors with VHL inactivation) AXL thus appears as a biomarker of resistance to anti-PD-1 therapy
CTLA-4 promoter gene methylation			
[39]	3 cohorts: 533 from TCGA database (Comprehensive methylation, expression, and immunogenomic data)	ccRCC	Mutiple therapies
	116	ccRCC	Not receiving ICI
	71	mccRCC (*n* = 67) mRCC (Other than ccRCC) (*n* = 4)	Anti-PD-1 second line (*n* = 46) Anti-PD-1 based combination (*n* = 25) CTLA4 promoter hypomethylation is an independent predictor of improved outcome (PFS and OS) in ICI-treated ccRCC CTLA4 gene promoter methylation thus appears as a biological biomarker of improved response to ICI combinations
Soluble molecules			
[40]	16	mRCC	Nivolumab On 507 molecules, 12 were significantly elevated in non-responders compared to responders After adjustment only RANKL was validated in patients with low RANKL levels had significant improvements in PFS RANKL thus appears as an independent biological biomarker of resistance to Nivolumab
Tumor content of T cells (T cell subsets, B cells, macrophages (and dendritic cells)			
[41]	24	mccRCC	Nivolumab + Ipilimumab Foxp3-CD4+ helper T cells, M1 macrophages, and CD137+ CD8+ T cells are potential predictive biomarkers of improved response Nivolumab + Ipilimumab
Levels of circulating soluble immune checkpoint-related proteins (BTLA, PD-1, PD-L1, PD-L2, CTLA-4, TIMP-3, LAG-3…)			
[42]	3 cohorts: 182 47 533 from TCGA	ccRCC (early and late stages)	No therapy or multiple therapies Soluble TIM-3 and soluble LAG-3 are associated with advanced disease Soluble PD-L2 is associated with recurrence Soluble BTLA and soluble TIM-3 are associated with decreased survival Soluble proteins may have prognostic value
Composition of the gut microbiota			
[43] Part of enrolled patients in the GETUG-AFU 26 NIVOREN microbiota translational substudy phase 2 trial	69 (vs. healthy volunteers)	mccRCC	Nivolumab after TKI Prior antibiotics (*n* = 11) No antibiotics (*n* = 58) Antibiotics compromised the clinical efficacy of ICBs Stool composition could be used to stratify the RCC patient population in responders and non-responders patients Antibiotics and TKIs shift the fecal microbiota composition Evidence for a favorable bacterial composition of feces and better clinical outcome to ICI
[44]	31	mRCC	Nivolumab (77%) Nivolumab + Ipilimumab (23%) Collection of stool prior to ICIs therapy Response to ICIs is characterized by changes in microbial species over the course of treatment Higher microbial diversity is associated with better treatment outcomes
[45] = MITRE Clinical Trial protocol (explore and validate a microbiome signature in a larger scale prospective study across several different cancer types, including RCC)	Recruiting	mRCC	Standard ICIs Primary outcome: measure the ability of the microbiome signature to predict 1 year (PFS) Secondary outcome: correlate the microbiome with toxicity and other efficacy end-points
[46] Evaluation of the impact of proton pump inhibitors (known to affect the microbiome) on ICIs clinical outcomes	2cohorts: 62 156	mRCC	Nivolumab + Ipilimumab (25 receiving proton pump inhibitors) Nivolumab (second- or third-line treatment) (88 receiving proton pump inhibitors) In neither cohort the concomitant use of proton pump inhibitors affected survival outcomes on ICIs
ccRCC molecular subgroups			
[47] BIONIKK Phase 2 Clinical Trial (primary endpoint: the objective response rate)	202	mRCC	Nivolumab (*n* = 61, groups ccrcc1 and 4) Nivolumab + Ipilimumab (*n* = 101, groups ccrcc1-4) VEGFR-TKI (*n* = 40, groups ccrcc2 and 3) Show the feasibility and positive effect of a prospective patient selection based on tumor molecular phenotype to choose between Nivolumab with or without Ipilimumab and a VEGFR-TKI in the first line treatment
[48] IMmotion150 Phase 2 Clinical Trial Previously untreated patients expressing PD-L1	305	mRCC	Atezolizumab (*n* = 103) Atezolizumab+ Bevacizumab (*n* = 101) Sunitinib (*n* = 101)
[49] IMmotion150 Phase 3 Clinical Trial The coprimary end points were PFS in patients with PD-L1+ disease and OS in the intention-to-treat population	915 (intention-to-treat population) 897 (safety population)	mRCC	PFS was improved for the combination treatment vs. Sunitinib Atezolizumab + Bevacizumab (*n* = 451) Sunitinib (*n* = 446) PFS was met at intermediate analysis No improvement in OS with the combination Improved median OS in patients with tumors characterized by T-effector/proliferative, proliferative, or small nucleolar RNA transcriptomic profiles = Insight into which patients may benefit from either therapy
Influence of the tumor microenvironment			
[50] Single-cell RNA sequencing	9	ccRCC (*n* = 7) Chromophe RCC (*n* = 2)	Insight into the possible cell of origin for the different RCC sub-types Tumor epithelia promote infiltration of immune cell, that may explain why ccRCC tumors respond to ICIs, (although they are known to have low neoantigen burden)
[51] Single-cell transcriptomic analysis (cancer and immune cells)	8 (primary or metastasis sites)	mccRCC (*n* = 7) Papillary RCC (*n* = 1)	Nivolumab (*n* = 2) Nivolumab + Ipilimumab (*n* = 1) Nivolumab + VEGF TKI (*n* = 2) No treatment (*n* = 3) Cells are reprogrammed after ICI treatment, and consequently the microenvironment, driving ultimately resistance to ICI Immune evasion is associated with PBRM1 mutation (reminiscent of what we described in Section 3.6) ICIs remodels the microenvironment and modifies the interplay between cancer and immune cells critical for response and resistance to ICIs

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
