# Peer review of "Biological Biomarkers of Response and Resistance to Immune Checkpoint Inhibitors in Renal Cell Carcinoma"

_cancers, 2023, doi:10.3390/cancers15123159_

Round 1
Reviewer 1 Report
summary
An interesting work. The authors give us an overview of the bibliography on biomarkers and molecular alterations related to resistance/response to immunotherapy in kidney cancer. Nevertheless, the work is not yet fully mature and can be greatly improved. It still lacks clarity, a concern about the title versus the content, a lack of figures and summary tables, missing references, small typos in to correct throughout. These concerns should be addressed before a more positive decision can be made, and ensure good dissemination of ideas, concepts, and readability for cancer readers.
My concerns are detailed below
Major concerns:
1-In my opinion, the title is not appropriate. There is little information and insight into the mechanistic aspects. There is also little proposal and discussion of new therapeutic approaches to consider.
The title is therefore misleading. The article focuses primarily on biomarkers of response and resistance, and that is fine. From beginning to end, the emphasis is on the need for biomarkers. The authors make this clear in the discussion and conclusions. Therefore, the authors can easily simplify their title by focusing on biomarkers. « Biomarkers of response and resistance to immunotherapy in RCC » or any similar title sounds appropriate to me. Make sure to differentiate from any similar work.
2- Another major concern is the lack of figures and tables
For the tables, I strongly recommend to add 1 or more tables with the following sinformations:
a-alteration/marker
b-the number of patients to have a better vision of the number of patients between studies (indeed often the authors say that studies must be confirmed on larger cohorts or that the number of patients is low, but it is difficult to compare between studies)
c-disease subtype or stage (localized/advanced/meta/recurrent ...)
d-treatment/combination ...(line of treatment if relevant)
e-if clinical trial, specify the clinical phase
f- main conclusion/finding
g- reference, or clinical trial number if relevant
3- As for the lack of figures, I'll leave it to the authors to decide whether or not it's paramount, or whether the tables can do the job, but if they want to highlight certain resistance mechanisms, it might make sense to add some, especially for genomic and epigenetic alterations that are hard to follow from the text of the current version alone. In this regard, They should take a look to the interesting review of M Moreira and colleagues: Cancer Drug Resist. 2020 Jul 2;3(3):454-471. doi: 10.20517/cdr.2020.16. eCollection 2020.Resistance to cancer immunotherapy in metastatic renal cell carcinoma
4- Also, I see a lot of key papers in the literature that are not cited. I make a non-exhaustive list of “hot” and “original” papers to help authors
Tumor and immune reprogramming during immunotherapy in advanced renal cell carcinoma
Bi, K; He, MX; (...); Van Allen, EM
May 10 2021 | May 2021 (Early Access) | 39 (5) , pp.649-+
PBRM1 loss defines a nonimmunogenic tumor phenotype associated with checkpoint inhibitor resistance in renal carcinoma
Liu, XD; Kong, W; (...); Jonasch, E May 1 2020 |11 (1)
Single-cell analyses of renal cell cancers reveal insights into tumor microenvironment, cell of origin, and therapy response
Zhang, YP; Narayanan, SP; (...); Chinnaiyan, AM Jun 15 2021 |118 (24)
Association of AXL and PD-L1 Expression with Clinical Outcomes in Patients with Advanced Renal Cell Carcinoma Treated with PD-1 Blockade
Terry, S; Dalban, C; (...); Chouaib, S Dec 15 2021 |27 (24) , pp.6749-6760
Integrative molecular characterization of sarcomatoid and rhabdoid renal cell carcinoma Bakouny, Z; Braun, DA; (...); Choueiri, TK
Feb 5 2021 |12 (1)
Association of Systemic Inflammation Index and Body Mass Index with Survival in Patients with Renal Cell Cancer Treated with Nivolumab
De Giorgi, U; Procopio, G; (...); Sternberg, CN
Jul 1 2019 | 25 (13) , pp.3839-3846
Tertiary lymphoid structures generate and propagate anti-tumor antibody-producing cells in renal cell cancer Meylan, M; Petitprez, F; (...); Fridman, WH
Mar 8 2022 | Mar 2022 (Early Access) | 55 (3) , pp.527-+
Adenosine 2A Receptor Blockade as an Immunotherapy for Treatment-Refractory Renal Cell Cancer Fong, L; Hotson, A; (...); Miller, RA Jan 2020 | 10 (1) , pp.40-53
Inhibition of FGFR Reactivates IFNg Signaling in Tumor Cells to Enhance the Combined Antitumor Activity of Lenvatinib with Anti-PD-1 Antibodies
Adachi, Y; Kamiyama, H; (...); Funahashi, Y Jan 15 2022 | 82 (2) , pp.292-306
5- I can't find a clear discussion of the PD-L1 marker. Although there are a few articles cited here and there, like in the article on RCC subtypes, it is still very evasive. I didn't really see a dedicated section, the authors could do a short section with main findings, ideas and concerns about this marker, with supporting references.
see
Alberto Carretero-González et al 2020
https://www.mdpi.com/2072-6694/12/7/1945
Keiichiro Mori et al 2021
https://www.sciencedirect.com/science/article/pii/S0302283820307831
6- Lack of bibliographic references in many sections of the text. It is important not to underestimate this aspect for a review. I give some examples
a-in introduction, the first sentence requires a citation to justify the given numbers
b-“…..ICIs targeting PD-1 and CTLA4 have been approved for RCC therapies…” This sentence is vague. Can you provide the name of the drug and the primary reference? Overall, this paragraph is vague. And contains no reference as well as the following
c-“…Tumor cells have developed several strategies to exploit these checkpoints and circumvent the host immune defenses. In other words, cancer cells are able to mimic the ligands of immune checkpoints to evade immune surveillance”
the end of the sentence requires a citation, and more globally the whole section
7- Many typos to correct I give some examples
a-Page 2 typo detected «last decade in advanced RCC with an overall survival reaching 47months”. Also add commas in the sentence to improve clarity
b- typo detected “or second-line therapy [11]. or in combination with pembrolizumab in the adjuvant setting”
c- typo detected page 3: “targets for immune therapy; These include 4-1BB” ; “ccRCC is known to be an immunogenic malignancy; Obviously,”
d- page 7 “but not in the TKI-treated group. However, among limitations od the study on Braun et al. [41] there”
e- page 8 “in these studies will need to be confirm”
f- page 8 “muciniphila abundance correlated with favorable outcomes while higher abundance of Hungatella hathewayi or Clostridium clostridioforme for ex-ample were increased in non-responders to nivolumab [53].” Add some commas to facilitate reading
g: “with ICIs and resistant to therapies; clinical trials are on-going to answer this question [57].” Replace semicolon by dot
h: in the conclusion part “Many works have still to be done in” To my knowledge, work does not take s
I: reference section : check the biblio format from 1 à 27
minor comments:
8- p6: “In both cohorts, CDKN2A genetic alterations were associated with poor response and survival in patients with urothelial carcinoma, but no association was..”
“poor response to what ?” Authos should clarify this statement
9- page 7 authors write a long paragraph: “ Interestingly, more recently, Chen et al. [43], using bioinformatic methods, identified 37 immune-related genes associated with PBRM1 mutations in ccRCC. In this original study, 520 ccRCC patients with survival information from TCGA (The Cancer Genome Atlas) database were analyzed………..….. However, as stated by authors, the biological functions, and molecular mechanisms of these genes in the immune response were not assessed and thus remain largely unknown. In addition, it is a retrospective, and not prospective study; thus, prospective clinical trials will be needed to validate the prediction performance of this model based on PBRM1 genetic status.”
I feel this paragraph is quite long, dealing with TCGA studies, mainly early cancers, not included in immunotherapy clinical trials, from a small bioinformatics study. Caution should be exercised. In fact, the authors themselves end up saying that these data should be taken with caution. I would reduce the size of this paragraph
10- similarly, there are other paragraphs that could be shortened, and why not by including these studies and main conclusions in the summary table
11- As much as possible when the authors talk about targeted therapy. It would be helpful to specify which ones (mTOR inhibitor, VEGFR, MET, others). It is not always defined in the current version
Reviewer 2 Report
Manuscript Title: Mechanism of Resistance to Immune Checkpoint Inhibitors and New Potential Therapeutic Approaches in Renal Cell Carcinoma
Manuscript ID: Cancers-2385902
. . . . . . . . . . . . . . . . . . . . . . . . . . . . . . . . . . . . . . . . . . . . . . . . . . . . . . . . . . . . . . . . . . . . . . . . . . . . . . . . . . . . . .
Comments on the current review article on the Mechanism of Resistance to Immune Checkpoint Inhibitors in Renal Cell Carcinoma and New Potential Therapeutic Approaches are presented below.
It would be appropriate to pay attention to some spelling rules. For example, "[11]. or in"
It would be appropriate to use TNFR instead of the abbreviation in this expression in "tumor necrosis factor receptor (TNRF)".
Authors should write the Latin binary names of living species in the text of the article, different from the general style of the text. For example, italicize. “Hungatella hathewayi”, “Clostridium clostridioforme”.
Adding appropriate figures and/or tables to this well-handled review article can reinforce the current manuscript in terms of quality.
Since there is confusion in the reference numbering, it may be suggested that the authors try to correct this again.
. . . . . . . . . . . . . . . . . . . . . . . . . . . . . . . . . . . . . . . . . . . . . . . . . . . . . . . . . . . . . . . . . . . . . . . . . . . . . . . . . . . . . .
Round 2
Reviewer 1 Report
This is a much improved version, with more in-depth discussions, added references, clarifications on several points and the addition of a table. Congratulations on their work and thank you for your detailed answers. I've just noted a few minor corrections to be made
- authors must ensure that the table is correctly referenced in the text (there was a typo on tqble in the main text) . i also recommend that they refer to it several times.
- authors should check the table for typos (examples on Nvolumab, mutiple therapies at )
- "After adjustment only RANKL was validated Patients..." add "in" after validated
- Authors should also be consistent with the terminology and start drug names with or without capital letters throughout the manuscript and table.
Reviewer 2 Report
Manuscript Title: Mechanism of Resistance to Immune Checkpoint Inhibitors and New Potential Therapeutic Approaches in Renal Cell Carcinoma
Manuscript ID: Cancers-2385902
R2
. . . . . . . . . . . . . . . . . . . . . . . . . . . . . . . . . . . . . . . . . . . . . . . . . . . . . . . . . . . . . . . . . . . . . . . . . . . . . . . . . . . . . .
The following are comments on the recent review article on the mechanism of resistance to immune checkpoint inhibitors in renal cell carcinoma and new potential therapeutic approaches.
It was noted that the authors have made most of the necessary corrections consistent with the comments.
. . . . . . . . . . . . . . . . . . . . . . . . . . . . . . . . . . . . . . . . . . . . . . . . . . . . . . . . . . . . . . . . . . . . . . . . . . . . . . . . . . . . . .
The manuscript needs revision on English spelling rules and sentence legibility.
